# Contribution of Influenza Viruses, Other Respiratory Viruses and Viral Co-Infections to Influenza-like Illness in Older Adults

**DOI:** 10.3390/v14040797

**Published:** 2022-04-12

**Authors:** Patricia Kaaijk, Niels Swaans, Alina M. Nicolaie, Jacob P. Bruin, Renée A. J. van Boxtel, Marit M. A. de Lange, Adam Meijer, Elisabeth A. M. Sanders, Marianne A. van Houten, Nynke Y. Rots, Willem Luytjes, Josine van Beek

**Affiliations:** 1Centre for Infectious Disease Control, National Institute for Public Health and the Environment (RIVM), 3721 MA Bilthoven, The Netherlands; patricia.kaaijk@rivm.nl (P.K.); nielsswaans@gmail.com (N.S.); alina.nicolaie@rivm.nl (A.M.N.); r.a.j.vanboxtel-4@umcutrecht.nl (R.A.J.v.B.); marit.de.lange@rivm.nl (M.M.A.d.L.); adam.meijer@rivm.nl (A.M.); lieke.sanders@rivm.nl (E.A.M.S.); nynke.rots@rivm.nl (N.Y.R.); willem.luytjes@rivm.nl (W.L.); 2Regional Laboratory for Public Health Kennemerland, 2035 RC Haarlem, The Netherlands; bruin1950@gmail.com; 3Department of Pediatric Immunology and Infectious Diseases, Wilhelmina Children’s Hospital, University Medical Center, 3584 EA Utrecht, The Netherlands; 4Spaarne Academy, Spaarne Gasthuis, 2133 TM Hoofddorp, The Netherlands; mvanhouten2@spaarnegasthuis.nl

**Keywords:** influenza virus infection, influenza-like illness, respiratory viruses, viral co-infections, viral interference, older adults

## Abstract

Influenza-like illness (ILI) can be caused by a range of respiratory viruses. The present study investigates the contribution of influenza and other respiratory viruses, the occurrence of viral co-infections, and the persistence of the viruses after ILI onset in older adults. During the influenza season 2014–2015, 2366 generally healthy community-dwelling older adults (≥60 years) were enrolled in the study. Viruses were identified by multiplex ligation–dependent probe-amplification assay in naso- and oropharyngeal swabs taken during acute ILI phase, and 2 and 8 weeks later. The ILI incidence was 10.7%, which did not differ between vaccinated and unvaccinated older adults; influenza virus was the most frequently detected virus (39.4%). Other viruses with significant contribution were: rhinovirus (17.3%), seasonal coronavirus (9.8%), respiratory syncytial virus (6.7%), and human metapneumovirus (6.3%). Co-infections of influenza virus with other viruses were rare. The frequency of ILI cases in older adults in this 2014–2015 season with low vaccine effectiveness was comparable to that of the 2012–2013 season with moderate vaccine efficacy. The low rate of viral co-infections observed, especially for influenza virus, suggests that influenza virus infection reduces the risk of simultaneous infection with other viruses. Viral persistence or viral co-infections did not affect the clinical outcome of ILI.

## 1. Introduction

Influenza is responsible for major morbidity and mortality burdens worldwide [1]. The elderly are at increased risk for severe disease and complications upon infection with influenza virus, and vaccination is considered important for prevention [1]. In the Netherlands, the annual seasonal influenza vaccination is offered to all older adults aged ≥60 years and to persons at high risk because of specific chronic comorbidities. Previously, we reported that influenza vaccination reduced the incidence of seasonal influenza virus infections in older adults, but did not reduce the overall incidence of influenza-like illness (ILI) cases in vaccinees [2]. The ILI incidence between vaccinated and nonvaccinated participants was similar, despite vaccination reducing the number of influenza virus infections. Apparently, other pathogens causing ILI are filling the gap. Influenza vaccination reduced the number of ILI cases caused by influenza virus infection by 73% and 51% in seasons 2011–2012 and 2012–2013, respectively. This, however, was offset by more ILI cases caused by other viruses, such as rhinovirus, seasonal coronavirus, human metapneumovirus (hMPV), or respiratory syncytial virus (RSV) [2]. Studies on viral etiology of ILI in community-dwelling older adults are scarce, and only a restricted number of viruses have been analyzed in these surveillance studies [3,4,5,6]. The aim of the present study was to investigate the contribution of influenza viruses, and other respiratory viruses to ILI in older adults (≥60 years) in an influenza season characterized by an influenza vaccine mismatch (2014–2015). For this purpose, naso- and oropharyngeal swabs from a cohort of 2366 older adults reporting ILI during the 2014–2015 influenza season were used to identify the respiratory viruses. In addition, the duration of the viral presence in the upper respiratory tract at 2 and 8 weeks after ILI onset was investigated. The impact of age, presence of having a chronic illness and/or clinical outcome of ILI on persistent virus was analyzed.

In children with respiratory illness, a high rate of virus/virus co-infection has been reported and has been associated with severity of disease [7,8,9]. Data on the occurrence of viral co-infections in older adults are scarce. To fill this gap in knowledge, we also assessed the occurrence of viral co-infections in older adults with ILI during two influenza seasons, i.e., 2012–2013 and 2014–2015.

## 2. Materials and Methods

### 2.1. Study Design

This prospective observational study in community-dwelling older adults, aged ≥60 years (60–94 years), was conducted from October 2014 through April 2015 in the Netherlands. Participants of the previous study (2012–2013) were re-invited and additional participants were recruited through the Civil Registry to reach the sample size of approximately 200 ILI cases [2]. Participants were part of the study for the entire duration of the season. The study was performed according to Good Clinical Practice, the Declaration of Helsinki and written informed consent was obtained from all participants. The study was approved by the ethical committee (http://www.trialregister.nl, accessed on 5 April 2022; NL4666).

The study design was similar to the previous studies [2]. Participants were instructed to report influenza-like illness (ILI) according to the Dutch Pel criteria, as defined by fever (≥37.8 °C) with at least one other symptom of coughing, headache, myalgia, sore throat, rhinitis, or chest pain [10]. A research nurse performed a home visit within 72 h of fever onset (acute phase) to collect naso- and oropharyngeal swab samples. A second and third visit was performed 2 weeks and 8 weeks (so-called “recovery visit”) later to investigate the duration of viral presence in the upper respiratory tract in this time frame. If a new ILI episode was reported, participants were visited again.

To compare the profile of the different viruses detected in ILI cases with that of participants without ILI, naso- and oropharyngeal swab samples were also collected from a group of asymptomatic participants at two sampling moments 14 days apart. To cover the whole period, a fixed number of participants aged ≥60 years and without ILI symptoms were invited every month of the study period up to a total of 200 asymptomatic persons, equally distributed over the different age groups of 60–64, 65–69, 70–74, 75–79 and >80 years. A consequence of this pragmatic approach was that the mean age as well as the vaccination coverage was slightly higher in this asymptomatic subset compared to the overall group (71.4 years with vaccination coverage of 80.5% versus 70.9 years and 68.2% vaccination coverage). Information on health, influenza vaccination status and demographics was recorded from all participants. The presence of respiratory symptoms and chronic illnesses was recorded in participants at home visits to collect swabs. To assess whether longer persistence of the detected virus(es) and/or the occurrence of viral co-infections affected the clinical outcome of ILI, it was monitored whether or not participants with ILI consulted a doctor for their ILI complaints.

### 2.2. Nasopharyngeal and Oropharyngeal Swabs

Naso- and oropharyngeal samples were obtained with a flocked nylon tipped swab and stored in modified liquid Amies transport medium (Eswab, Copan, Brescia, Italy) at −80 °C within <8 h after sampling.

### 2.3. Analysis of Viruses by MLPA

Viruses were detected by analysis of DNA and RNA isolated from the swabs by Multiplex Ligation-dependent Probe Amplification (MLPA) assay (RespiFinder Smart 22 kit (Pathofinder, Maastricht, The Netherlands)), as described before [2]. Influenza virus-positive samples were subtyped by real-time RT-PCR using the Roche LightCycler 480 system with slightly modified protocols [11,12]. The gene coding for HA of influenza A(H3N2) viruses were Sanger-sequenced using universal influenza virus type A primers (available on request) directly from the clinical specimens. HA sequences were used to identify the phylogenetic clade to which the viruses belong. As reference for clade designation following phylogenetic analysis, guidance of WHO CC, London, UK is used (https://www.crick.ac.uk/partnerships/worldwide-influenza-centre/annual-and-interim-reports, accessed on 5 April 2022).

### 2.4. Statistical Analysis

In order to compare baseline characteristics, such as sex, vaccination status, chronic illness or persistent virus, and having ILI between the different groups, Pearson’s χ^2^ testing was applied. Differences in age were assessed using independent samples *t* test of the means, and the male/female distribution in infections with a given pathogen or presence of persistent virus was evaluated using the two-way Fisher’s exact test.

The level of significance (*p* value) was set to 0.05. These statistical analyses were performed using IBM SPSS Statistics version 24.0.0.1 (IBM Corporation, Armonk, NY, USA).

Vaccine-effectiveness (VE) was determined by test-negative design analysis of ILI positive participants in the influenza-active period in the Netherlands in 2014–2015 as previously described [2,13]. The VE is calculated as (1–odds ratio [OR]) × 100% with 95% confidence interval (CI) and is calculated per influenza virus subtype or lineage. Period in the season (early and late season), sex, smoking, chronic illness, and age were regarded as potential confounders and their association with influenza virus positivity was analyzed with univariate logistic regression [14]. Variables with *p* value < 0.20 were considered in the multivariable analysis. VE analysis was performed with SAS version 9.4 software.

In order to analyze the presence and frequency of viral co-infections at the genus level in the cohort, all virus variables were assessed in sets of two-virus combinations with the two-way Fisher’s exact test that was adjusted for multiple testing using the Benjamini-Hochberg procedure. Results of these analyses are presented as the OR of the odds of being infected with pathogen A when already infected with pathogen B compared to the odds of being infected with a singular pathogen A. The 95% CI is used to estimate the precision of the OR. Adjusted *p*-values (*p*_adj_) were calculated after multiple testing correction, with the false discovery rate (FDR) set to 10%. To determine the frequencies of three-virus combinations, two-virus combinations with a significant result (*p*_adj_ < 0.10) were tested against remaining pathogens using the two-way Fisher’s exact test as described above. Statistical analysis for viral co-infections was performed using R version 3.4.3 (www.r-project.org, accessed on 5 April 2022).

## 3. Results

### 3.1. Study Cohort and ILI Incidence

In this prospective study, data were collected from influenza season 2014–2015 (from October 2014 to April 2015), a season with a vaccine mismatch [15,16]. From season 2012–2013, a longer influenza season than average, data on single respiratory infections were previously described. In this study we also included unpublished data on viral co-infections from 2012–2013 for comparison with season 2014–2015 [2]. A flow diagram for participants included in 2014–2015 is presented (Figure 1), the flow diagram of the seasonal cohort 2012–2013 was published previously [2]. A total of 2366 older adults from season 2014–2015 were included for statistical analysis, i.e., persons with ILI (*n* = 252), persons without ILI (2114). Two persons presented with two ILI episodes with a fever-free period of at least 2 weeks in between, resulting in two more acute ILI samples (*n* = 254). From the persons without ILI, a group of asymptomatic controls was selected (*n* = 205). This allowed us to compare the symptomatic group with an asymptomatic group with viral presence as carriage without ILI symptoms. The overall ILI incidence of season 2014–2015 was 10.7% (252/2366) and the influenza vaccination coverage was 68.2% (Table 1). The overall ILI incidence was not significantly different between vaccinated (10.4% (168/1614)) and unvaccinated persons (11.2% (84/752)). The average age of vaccinated individuals was significantly higher than the age of unvaccinated individuals (72.0 versus 68.5 years; *p* < 0.0001), and the average age of participants without ILI was slightly higher than ILI cases (71.1 versus 69.6 years; *p* = 0.001). Participants with chronic illness (i.e., having cardiovascular disease, auto-immunity, diabetes, chronic respiratory conditions and/or malignancy) were vaccinated significantly more often than participants without chronic illness (ILI cases: 79.6% versus 56.1%; *p* < 0.0001; asymptomatic controls: 88.2% versus 75%, *p* = 0.02). The percentage of participants with chronic illness was comparable in the ILI and asymptomatic group (Table 2).

### 3.2. Influenza Virus Infection in ILI Cases and Influenza Vaccine Effectiveness

The 2014/2015 influenza vaccine contained A/California/7/2009 (H1N1)pdm09-like virus, A/Texas/50/2012 (H3N2 clade 3c.1), and B/Massachusetts/2/2012-like virus (Yamagata lineage) strains. Influenza viruses were detected in 39.4% (100/254) of the acute ILI samples (Figure 2, Appendix A). Of the influenza viruses, 76% were of type A, of which 10.5% was subtype A(H1N1)pdm09 and 89.5% A(H3N2), and 24% were of type B, all belonged to the B/Yamagata lineage (Appendix A). The circulating H3N2 strains, i.e., A(H3N2) clade 3C.2a and clade 3C.3b, mismatched with the vaccine A/Texas/50/2012 (H3N2 clade 3c.1) strain [10,11]. Males were more often infected with influenza viruses than females during acute ILI phase: 47% versus 33% (*p* = 0.028), while vaccination status and age did not differ among male and female ILI cases. No sex preference was found for any specific influenza virus subtype.

Next, we evaluated whether influenza vaccination reduced the percentages of influenza virus infections or ILI cases. The percentages of overall influenza virus infections (Table 3) and ILI cases (Table 1) were not lower in vaccinated compared to unvaccinated individuals.

After correcting the data for potential confounders, the overall adjusted VE for all influenza vaccine strains of season 2014/2015 was −1% (95% CI, −88% to 46%). The VE for predominant influenza virus subtype A (H3N2) strain was −26% (95% CI, −161% to 39%), whereas the VE was highest for influenza B/Yamagata-like strain, i.e., 49% (95% CI, −39% to 81%). However, the point estimates of VE were not statistically significant. (Table 4). Additional sensitivity analyses for multiple ILIs, households with ILI, and the presence or absence of other virus infections did not affect data on vaccine effectiveness (data not shown).

### 3.3. Other Respiratory Viruses Detected in ILI Cases from Season 2014–2015

For the influenza season 2014–2015, in 78.7% (200/254) of the acute ILI samples, at least one respiratory virus was identified (Figure 2A; Appendix A). Apart from influenza virus (39.4%), other viruses were detected with substantial contribution to ILI cases. We only considered viruses with contribution to ILI cases of >5% here, which were human rhinoviruses (17.3%), seasonal coronaviruses (9.8%; of which 48% OC43, 36% 229E, 8% NL63, and 8% HKU1), respiratory syncytial virus (RSV) (6.7%), and human metapneumovirus (hMPV) (6.3%).

### 3.4. Persistence of Viruses in ILI Cases

The percentage of ILI samples in which at least one respiratory virus was identified was reduced from 78.7% at the acute phase to 28.7% at 14 days after ILI onset (Appendix A). Nevertheless, DNA/RNA of the same respiratory virus was detected in 19.3% (49/254) of the samples at 14 days. At the recovery time point, at 8 weeks after ILI onset, in 14.6% of the samples a respiratory virus was detected (Appendix A, Figure 2A), but in only 3% (8/254) of these ILI samples the same virus was detected as at ILI onset. In comparison, in the asymptomatic controls in 13.7% of the first samples, and in 14.1% of the samples taken 14 days later, respiratory viruses were detected (Appendix A). In 3% (6/205) of the ILI samples obtained from asymptomatic controls, the same virus was detected at both sampling time points.

In 11% (11/100) of the ILI cases with influenza virus infection, the same influenza virus strain was still detectable 14 days after ILI onset. This was observed in nine male and two female cases. In addition, nine out of the 11 cases where persistent influenza virus was detected had underlying chronic illness. Strikingly, six out of these nine cases had underlying chronic respiratory conditions, of which two also had cardiovascular disease, and another two cases had cardiovascular disease. Longer persistence of influenza virus in the respiratory tract was not associated with a more serious course of ILI, as no physician was consulted because of any ILI complaints by the participants with persistent influenza virus detected. Influenza viruses were detected in only 1.2% of the recovery samples collected at 8 weeks post ILI onset, which is comparable to the percentage of influenza positive samples of asymptomatic controls (0.5% and 1% in the two separate sample collections).

Other respiratory viruses that persisted for 14 days after ILI onset were found in 47.7% (21/44) of the rhinovirus cases, in 24% (6/25) of the seasonal coronavirus cases, in 23.5% (4/17) of the RSV cases and in 18.8% (3/16) of the cases where hMPV was detected at ILI onset. These viruses were present in lower frequency in recovery and asymptomatic control samples, although rhinoviruses and coronaviruses were still regularly detected in recovery samples of the ILI cases (respectively, 6.3% and 5.1%) and in the two samples of the asymptomatic group (respectively, 8.3%/5.9% and 3.9%/2.4%) (Figure 2, Appendix A). Only rhinoviruses were detected in both ILI and recovery samples of the same individual, in 9.1% (4/44) of the cases. However, rhinoviruses were not subtyped and therefore it cannot be excluded that these were of different subtypes and thus could be new infections rather than persistent infections.

The same viruses, including influenza viruses, were more often detected 14 days after ILI onset in participants with underlying chronic illness (*p* = 0.012). However, no severe infectious disease was reported in any of the ILI cases, and none of the participants with persistent virus(es) detected 2 weeks after ILI onset consulted a physician for ILI. In addition, no statistical significant differences in age (*p* = 0.17) or sex (*p* = 0.20) were observed between those with persistent virus(es) detected and those without.

### 3.5. Viral Co-Infections

In season 2014–2015, in 4.8% (12/254 of the samples from the acute ILI phase) more than one respiratory virus was detected in the nasopharyngeal and/or oropharyngeal swabs (Appendix A). In one ILI case more than two viruses were detected, i.e., influenza virus, seasonal coronavirus and RSV. There were no hospital admissions among the ILI cases with viral co-infections, and in this group of community-dwelling older adults the general practitioner was not consulted for ILI complaints, supportive of a relatively mild clinical course of ILI.

A calculation was made of how often a specific co-infection could be expected based on the frequency of the individual viruses detected per season. We assumed independence of the occurrence of the individual viruses when a joint occurrence of two viruses was observed. During ILI episodes, the co-infections of influenza virus on the one hand and rhinovirus, coronavirus, hMPV or parainfluenza virus on the other hand appeared to occur less often than expected if occurrences of these co-infections were random based on the frequency of the detected single pathogens (*p*_adj_ < 0.10) (Figure 3).

For comparison, the occurrence of viral co-infections in 2012–2013 was studied as well. Data from season 2011–2012 were not considered here as it was a mild influenza season with a small number of influenza cases and with consequently low statistical power for the analyses on viral co-infections. In 7.6% (21/275) of the acute ILI samples from season 2012–2013, viral co-infections were detected. In line with the findings from season 2014–2015, in 2012–2013 co-infections of influenza virus with rhinovirus, coronavirus, hMPV or parainfluenza virus were observed less often than expected if occurrence of co-infections were random (*p*_adj_ < 0.10). In addition, co-infections of influenza virus with RSV, and co-infections with rhinovirus and coronavirus, occurred less frequently than expected based on the frequency of these single pathogens detected in swabs from ILI cases (*p*_adj_ < 0.10) (Figure 3).

## 4. Discussion

In this prospective observational study performed in 2014–2015, in 78.7% of the acute influenza-like illness (ILI) samples at least one potential respiratory virus was identified. Influenza virus was involved in 39.4% of the ILI cases, comparable to the 34.2% observed in the study performed with this cohort of older adults (aged 60–94 years) in season 2012–2013, a season with moderate vaccine effectiveness [2]. Where the 2012–2013 season was the longest influenza epidemic of the previous 20 years, the season of a year earlier (2011–2012) proved to be mild. In 2011–2012, only 18.9% of the reported ILI cases were caused by influenza virus [2]. Our data are consistent with a recent literature and meta-analysis that found that laboratory-confirmed influenza accounted for approximately one third of all acute respiratory infections for which medical care was sought during influenza seasons (2004–2017) in the European region [17].

Apart from influenza virus, other respiratory viruses were also detected in the samples taken during acute ILI, i.e., rhinovirus (17.3%), seasonal coronavirus (9.8%), RSV (6.7%) and hMPV (6.3%) were detected. Most of these viruses were cleared 14 days after ILI onset, i.e., in approximately 80% of the cases. The same viruses were more often detected 14 days after ILI onset in participants with underlying chronic illness (*p* = 0.012). However, no severe infectious disease was reported in any of the ILI cases, and none of the participants with persistent viruses detected 2 or 8 weeks after ILI onset consulted a physician for ILI. No differences in age or sex were observed between ILI cases with and without persistent virus(es) detected. Only rhinoviruses were detected in both samples from the same individual taken during acute ILI and 8 weeks later. This could indicate that rhinovirus persisted longer, but we cannot exclude the occurrence of new infections because data on molecular subtyping of rhinovirus were not available. Moreover, rhinovirus was detected in 8.3% and 5.9% of the two samples taken from asymptomatic controls as well, indicating frequent occurrence of rhinovirus in this time of the season, even without ILI complaints. In this study, it is good to consider that for determining the viral etiology of ILI the presence of viral DNA/RNA was assessed instead of replicating virus.

ILI incidence of season 2014–2015 was 10.7% (252/2366) and vaccination coverage was 68.2%, in comparison in season 2011–2012 and in 2012–2013 ILI incidence was, respectively, 7.2% (143/1992) and 11.6% (275/2368), and vaccination coverage was 75.9% and 68.5% [2]. The number of influenza virus infections was not affected by vaccination in this season; the overall adjusted VE for all influenza vaccine strains of season 2014–2015 was low and the point value estimate of VE was not statistically significant (VE of −1% (95% CI, −88% to 46%)). Low VE against influenza A(H3N2) infection due to vaccine mismatch was also reported elsewhere in Europe in 2014–2015 and has been attributed to circulation of a drifted influenza A(H3N2) strain [15,16,18]. For the seasons 2011–2012 and in 2012–2013, with, respectively, high and moderate vaccine effectiveness (VE of 73% and 51%), we showed that although vaccination reduced the number of influenza virus infections, the overall number of ILI episodes, regardless of the pathogen causing it, was not reduced [2]. In line with this, we show here that, also in an influenza season with vaccine mismatch, vaccination did not affect the total number of ILI cases in a cohort of older adults.

Studies in children showed high rate of viral/viral co-infections (18−34%) [7,8,9,19], although it should be taken into account that children included in these studies were hospitalized for respiratory infections, which could explain the high percentage of co-infections. In these children, the occurrence of co-infections was associated with worse prognosis. Data on viral co-infections in the older adult population are scarce. We therefore investigated the occurrence of viral co-infections in older adults in the influenza seasons 2012–2013 and 2014–2015. In 7.6% (2012–2013) and 4.8% (2014–2015) of the ILI cases more than one respiratory virus was detected, which is considerably lower than has been reported from studies in hospitalized children [7,8,9]. In our study, the occurrence of viral co-infections in older adults with ILI did not affect the clinical outcome; none of the participants with viral co-infections visited a physician. Interestingly, it seems that co-infections did not occur randomly: infections of influenza virus together with rhinovirus, seasonal coronavirus, hMPV or parainfluenza virus were observed less frequently than expected based on the frequency of the detected single viruses. These findings are in agreement with a recent study that was based on virological data of 44,230 episodes of respiratory illness over a 9-year time frame, that provided strong statistical evidence for the existence of a negative interaction between seasonal influenza A virus and rhinovirus, at both the population and individual host scale [20]. Additionally, in another study, a lower prevalence of co-infection with influenza virus was found in rhinovirus-positive children compared to rhinovirus-negative children without symptoms of severe pneumonia [21].

The reduced frequency of certain combinations of viruses may indicate the occurrence of viral interference. Viral interference is a form of resistance that may occur after a host becomes infected with one virus, and prevents infection and replication of a second virus. It has been speculated that upon infection by the first virus, either through the cell-mediated response, or more likely, through the innate immune response, a rapid state of immune activation that protects against simultaneous infections by other viruses is induced causing a temporarily ‘antiviral state’ [20,22,23,24]. Several epidemiological studies have supported the occurrence of viral interference as well, in particular during the 2009 influenza pandemic [17,18,19,20]. Although, it should be considered that complex interactions exist between viruses, populations, and the environment [25].

In agreement with our finding that co-infections of influenza virus with rhinoviruses occurred less frequently than expected, several studies indicated that rhinovirus may have inhibited the circulation of A(H1N1)pdm09 virus in Norway and France [26,27]. In our study, in season 2012–2013, but not 2014–2015, influenza virus infections were also found to occur less frequently in combination with RSV. In line with this finding, early epidemics of influenza virus infection have been observed to delay RSV epidemics. In France, RSV emerged late at the end of December 2009, when A(H1N1)pdm09 started to decline, which was an unusual pattern compared with previous years [28]. Additionally, in Israel, RSV infections were delayed and started after A(H1N1)pdm09 had declined [29].

In an in vitro study using a mathematic model, it was found that a fast-growing virus may reduce replication of slow-growing viruses during a co-infection [30]. Interestingly, these epidemiological findings match with the replication parameters for these respiratory viruses. The growth rate of rhinovirus is higher than that of influenza virus, whereas the infection time is shorter, i.e., it takes less time for a newly produced infectious rhinovirus particle to infect a susceptible cell compared to an influenza virus particle [30]. This may explain why rhinovirus may have inhibited influenza virus infections. On the other hand, the growth rate of influenza virus is higher than that of RSV and has a shorter infection time [30], which may explain why the influenza pandemic in turn delayed the RSV epidemic. In light of the coronavirus disease−2019 (COVID-19) pandemic, it might be interesting to notice that in our study co-infections with seasonal coronavirus and influenza virus, and seasonal coronavirus with rhinovirus, occurred less often than expected, which may be indicative of the occurrence of viral interference.

As in our previous study [2], a limitation of the present study is that, due to our method of recruitment, the more frail institutionalized elderly are not included. Consequently, our data acquired from the generally healthy community-dwelling older adults do not necessarily apply to the group of more frail elderly, in which viral exposure and susceptibility to infections may be different. Furthermore, the Dutch Pel criteria (i.e., fever (≥37.8 °C) with at least one other symptom of coughing, headache, myalgia, sore throat, rhinitis, or chest pain) were used here for the definition of ILI [10]. It can be speculated that the specific ILI definition used in this study, although most other ILI definitions also include fever and cough [31,32,33], may have affected the number and type of pathogens that were detected.

## 5. Conclusions

In line with our previous studies [2], data of this study show that influenza virus is the most frequently detected pathogen in older adults presenting with ILI. Furthermore, we show that, also in an influenza season with vaccine mismatch, vaccination did not affect the total number of ILI cases. It is tempting to speculate that viral interference by influenza virus may be involved here. When vaccination is effective and influenza virus infection is prevented, viral interference by influenza virus will not take place and other viruses may therefore have more opportunities to infect and cause ILI. In this cohort of older adults, the percentage of co-infections of influenza virus together with other viruses was remarkably low.

## Figures and Tables

**Figure 1 viruses-14-00797-f001:**
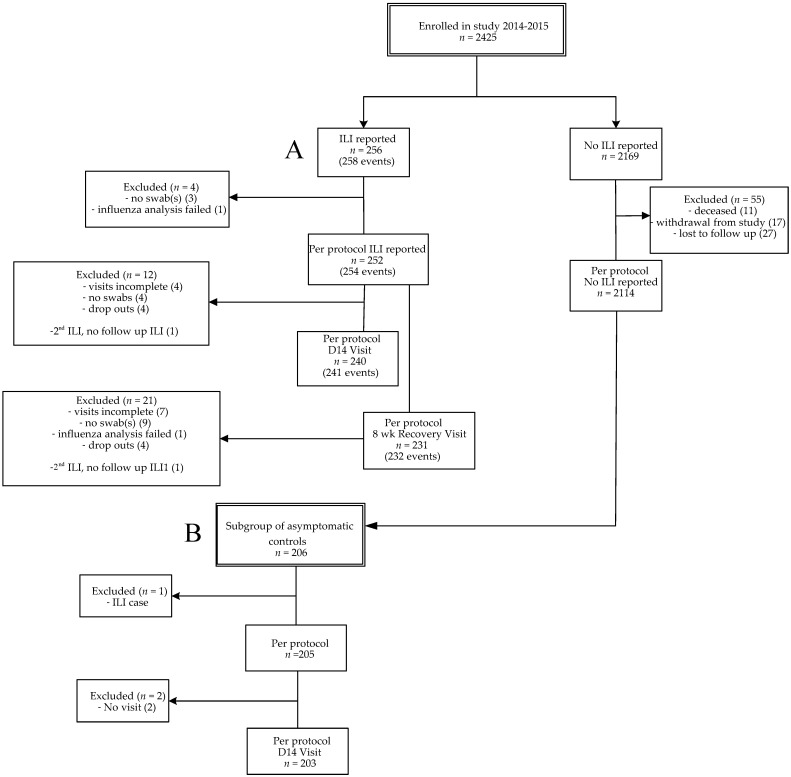
Flow diagram of enrollments. Influenza-like illness (ILI) cases (2014–2015) (**A**) and the subgroup of asymptomatic controls (**B**). A subject could have multiple ILI episodes per season. Per protocol was defined when the sample was taken <72 h after start of fever. For the recovery visit, the window was 7–9 weeks after ILI onset. Subjects were considered lost to follow-up if they did not respond to the end of study mailing and had no ILI visit. Every month of the study period a fixed number of asymptomatic participants, equally distributed over the different age groups, were invited for swab sampling.

**Figure 2 viruses-14-00797-f002:**
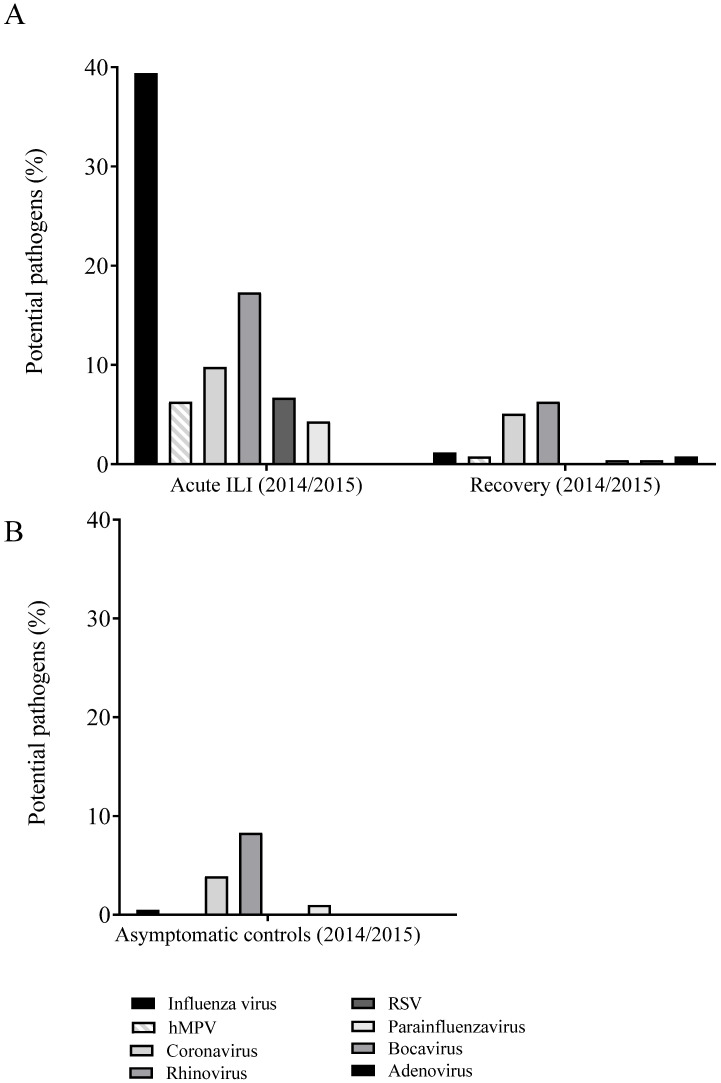
Incidence per virus that were detected in naso- and oropharyngeal swabs of influenza-like illness (ILI) cases in the acute phase (left panel) and at recovery (i.e., 8 weeks later) (right panel) (**A**) and of first samples of asymptomatic controls, i.e., participants aged ≥60 years, and without ILI symptoms (**B**) in influenza season 2014–2015. The percentages were calculated per ILI event. Multiple pathogens could be detected in a single event and therefore contribute to the incidence for multiple pathogens. Abbreviations: hMPV, human metapneumovirus; ILI, influenza-like illness; RSV, respiratory syncytial virus.

**Figure 3 viruses-14-00797-f003:**
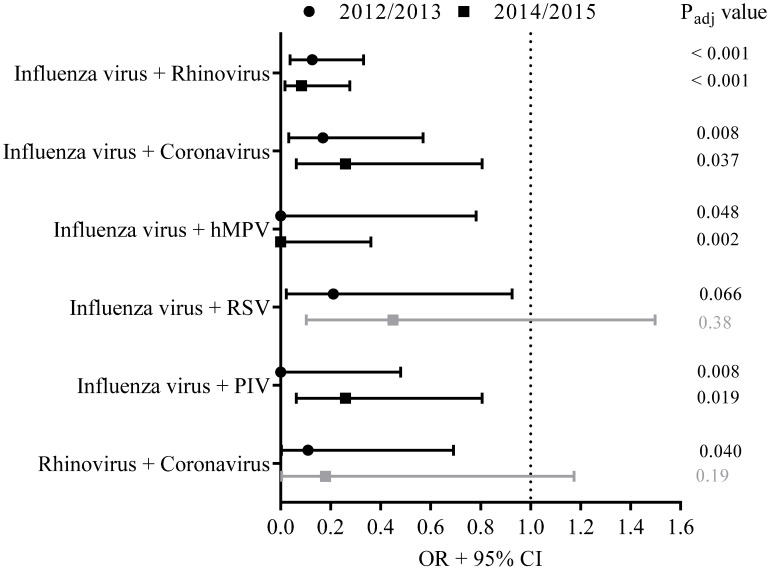
A calculation was made whether a specific virus co-infection appeared to occur less often than expected based on the frequency of the detected single pathogens in season 2014–2015 (squares). For comparison, the occurrence of viral co-infections in 2012–2013 was studied as well (circles). Results of these analyses are presented as the OR of the odds of being infected with pathogen A when already infected with pathogen B compared to the odds of being infected with a singular pathogen A. The 95% CI, presented as bars, was used to estimate the precision of the OR. Adjusted *p*-values (*p*_adj_) were calculated, and *p*_adj_ < 0.10 was considered as significantly different (indicated with black lines), *p*_adj_ > 0.10 was considered as not significantly different (indicated with grey lines).

**Table 1 viruses-14-00797-t001:** Demographic characteristics of participants.

2014/2015						
	All (*n* = 2366)	ILI (*n* = 252)	No ILI (*n* = 2114)	Asymptomatic Controls (*n* = 205)	*p* Value ILI vs. No ILI	*p* Value ILI vs. Asymptomic Controls
Male sex	1205 (50.9%)	121 (48.0%)	1084 (51.3%)	107 (52.2%)	NS	NS
Age, y, mean (range)	70.9 (60–94)	69.6 (60–88)	71.1 (60–94)	71.4 (60–88)	0.001	0.001
Influenza vaccination 2014/2015	1614 (68.2%)	168 (66.7%)	1446 (68.4%)	165 (80.5%)	NS	0.001

Data are presented as No. (%). Abbreviations: ILI, influenza-like illness; No ILI, without symptoms of influenza-like illness; NS, not significant.

**Table 2 viruses-14-00797-t002:** Occurrence of chronic illness in combination with vaccination status.

	ILI (*n* = 252)	Asymptomatic Controls (*n* = 205)	*p* Value
Any chronic illness *	113 (44.8%)	85 (41.5%)	NS
% vaccinated with any chronic illness	90 (79.6%)	75 (88.2%)	
% vaccinated without chronic illness	78 (56.1%)	90 (75%)	
*p* value (Pearson χ^2^ test)	0.0001	0.02	

Data are presented as No. (%). Abbreviations: ILI, influenza-like illness; NS, not significant. * Participants in this study with chronic illness had cardiovascular disease, auto-immunity, diabetes, chronic respiratory conditions and/or malignancy.

**Table 3 viruses-14-00797-t003:** Pathogens detected in participants with acute ILI relative to vaccination status in season 2014/2015.

	Vaccinated	Non-Vaccinated	*p* Value
	(*n* = 168 Participants)	(*n* = 84 Participants)	
Influenza virus	69 (41.1%)	31 (36.9%)	NS
Influenza virus A	57 (33.9%)	19 (22.6%)	NS
A(H3N2)	52 (31.0%)	16 (19.0%)	NS
- 3C.2a	31 (18.4%)	8 (9.5%)	NS
- 3C.3b	14 (8.3%)	8 (9.5%)	NS
A(H1N1)pdm09	5 (3.0%)	3 (3.6%)	NS
Influenza virus B	12 (7.1%)	12 (14.3%)	NS
- Yamagata-like			
Coronavirus	16 (9.5%)	9 (10.7%)	NS
hMPV	13 (7.7%)	3 (3.6%)	NS
RSV	9 (5.4%)	8 (9.5%)	NS
Rhinoviruses	28 (16.7%)	16 (19.0%)	NS
Parainfluenza virus	6 (3.6%)	5 (6.0%)	NS

Data are presented as No. (%). Abbreviations: hMPV, human metapneumovirus; RSV, respiratory syncytial virus; NS, not significant (*p* value > 0.05).

**Table 4 viruses-14-00797-t004:** Vaccine effectiveness during influenza active period 2014–2015.

			*n*	Odds Ratio (95% CI)	VE (95% CI)
2014/2015					
Influenza virus			210	1.005 [0.538–1.878]	−1% [−88–46%]
	A		186	1.170 [0.571–2.399]	−17% [−140–43%]
		A(H3N2)	178	1.261 [0.609–2.610]	−26% [−161–39%]
		- 3C.2a	149	1.965 [0.733–5.270]	−96% [−427–27%]
		- 3C.3b	132	0.400 [0.131–1.222]	60% [−22–87%]
		A(H1N1)pdm09	118	1.339 [0.274–6.555]	−34% [−555–73%]
	B	B/Yamagata-like	134	0.509 [0.187–1.389]	49% [−39–81%]

Abbreviations: CI, confidence interval; VE, vaccine effectiveness.

## Data Availability

The raw data supporting the conclusions of this article will be made available by the authors upon request, with consideration of the participants’ privacy and ethical rights.

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
