# Peer review of "Contribution of Influenza Viruses, Other Respiratory Viruses and Viral Co-Infections to Influenza-like Illness in Older Adults"

_viruses, 2022, doi:10.3390/v14040797_

Round 1
Reviewer 1 Report
Below is a list of my comments.
The methods should be better described (especially the characteristics of the patients).
The results should be better described so that they can be compared with the information contained in the discussion.
The discussion should be significantly shortened.
Methods:
1. The population of infected people should be characterized in more details, especially age groups (,,≥ 60 years” may be not enough when discussing the results), ad additionally the range of age in Table 1 is: (60-94). There may be differences in the results for people aged 60 and 94.
86 line: ,,200 asymptomatic persons, equally distributed over the different age groups” – what groups are these?
- “chronic illness” should be defined, it means coexisting diseases?
Line 90
,,Information on health, influenza vaccination status and demographics was recorded from all participants,, - these collected data should be presented in the Results.
Line 42. Introduction
,,Previously, we reported that influenza vaccination reduced the incidence of seasonal influenza virus infections in older adults, but did not reduce the overall incidence of influenza-like illness (ILI) cases in vaccinees,,
Please explain why influenza vaccine was expected to reduce ILI?
What is the definition of ILI?
Similarly, please note other points in the discussion:
Line 299 Discussion
,,vaccination against influenza virus had no influence on the frequency of total cases of ILI in community-dwelling older adults,,
Line 345
,,vaccination reduced the number of influenza virus infections, the 345 overall number of ILI episodes was not reduced,,
Line 337
,,vaccination did not affect the total number of ILI cases in a cohort of older adults,,
Line 54 and 193
,,Influenza vaccination reduced the number of ILI cases caused by influenza virus infection ,,
It is not clear when the authors write about cases of influenza, cases of viral infections of the upper respiratory tract and cases of viral infections of the upper respiratory tract caused by viruses other than influenza.
Results
Table 1 should be better described, what does ,,no ILI” mean?
Table 2 ,,any chronic illness,, –
What does ,,chronic illness,, mean? Maybe comorbidities?
Line 194
,,The percentages of overall influenza virus infections (Table 3) and ILI cases (Table 1) were not lower in vaccinated compared to unvaccinated individuals,,
Why is the impact of influenza vaccination on ILI was assessed?
Conclusions
Line 416
,,vaccination did not affect the total number of ILI cases,, – The same. Why is the impact of influenza vaccination on ILI was assessed?
Figure 2B - Should be better explained
Line 222
,,viruses that were detected in >5% of the ILI cases were human rhinoviruses (17.3%),,
Please, specify how much is > 5%.
line 236
,,of the influenza virus-infected ILI cases,, - means influenza or influenza virus infection? Please, specify.
Line 324
,,No differences in age or sex were observed between …,,
This has not been shown in detail in the Results.
,,…ILI cases with and without persistent virus(es) detected,,
This statement has not been sufficiently proven, there is no data in the results on the basis of which this observation was made.
Line 338
,,The number of influenza virus infections was not affected by vaccination in this…,,
Rather, in this study… are there not enough samples observed to reach such a conclusion?
Line 394
,,The growth rate of rhinovirus is higher than that of influenza virus, whereas the infection time is shorter, i.e. it takes less time for a newly produced infectious rhinovirus particle to infect a susceptible cell compared to an influenza virus particle [31]. This may explain why rhinovirus may have inhibited influenza virus infections. On the other hand, the growth rate of influenza virus 398 is higher than that of RSV and has a shorter infection time [31],,
It should be supplemented that the results were obtained in vitro using mathematical modelling.
line 400
,,In light of the novel corona virus disease 2019 (COVID-19) pandemic, it might be interesting to notice that in our study also co-infections with coronavirus and influenza virus, and coronavirus with rhinovirus occurred less often than expected, which may be indicative for the occurrence of viral interference,,
The data was published?
,,…novel coronavirus disease 2019 (COVID-19) pandemic,, - COVID-19 pandemic or Coronavirus Disease 2019 (COVID-19) pandemic.
Author Response
"Please see the attachment."

Reviewer 2 Report
This very interesting and relevant paper looks at the important question of incidence of respiratory viruses (influenza, RSV, seasonal coronaviruses, hMPV) in an elderly population during an influenza season with a mismatched influenza vaccine. The goal of this study is to investigate the contribution of the different viruses to ILI in this population. The authors present findings on low rates of co-infection of viruses and speculate on the ramifications of influenza vaccine efficacy for other ILI causing viral infections. Additionally of particular interest, given the possibility of Omicron SARS-CoV-2 from a chronically infected population, is the evidence of persistence of respiratory virus infections in participants with chronic illnesses.
Minor comments:
- While taking into account issues of privacy, it would be helpful to the reader to list what types of chronic illnesses were present in the population, especially in the patients who had persistent infections.
- Is there information on specific seasonal coronaviruses? If not, a follow up study on these samples providing data on specific seasonal coronaviruses would be of great interest to the scientific community.
- Line 78: "A second and third visit were performed 2 and 8 weeks later to investigate the duration of viral presence in the upper respiratory tract in this time frame." Later in the paper these are referred to as "recovery" visits. It would help to use this nomenclature in the materials and methods. It would also help to define recovery as 8 week post ILI sample for the caption in Figure 2A to distinguish from 2 week recovery.
- Line 229-230: "At the recovery time point, at 8 weeks after ILI 229 onset, in still 14.6% of the samples a respiratory virus was detected" should be " A question here is whether you mean to imply by using the word "still" that this is residual nucleic acid from the original virus. If this is not implied, then remove the word "still". If keeping the word, correct grammar is "still detected".
- Line 315: "In only, 1% of the samples" remove comma.
- Line 317: "Apart from influenza virus, also other respiratory viruses were detected" should be "other respiratory viruses were also detected"
Author Response
"Please see the attachment."

Round 2
Reviewer 1 Report
Dear Authors,
Thank you for response to all my comments.